# Proteome allocation is linked to transcriptional regulation through a modularized transcriptome

Arjun Patel[1], Dominic McGrosso[2], Ying Hefner[1], Anaamika Campeau[2], Anand V. Sastry[1], Svetlana Maurya[2], Kevin Rychel [1], David J. Gonzalez[2,3] & Bernhard O. Palsson [1,4] ✉

It has proved challenging to quantitatively relate the proteome to the transcriptome on a per-gene basis. Recent advances in data analytics have enabled a biologically meaningful modularization of the bacterial transcriptome. We thus investigate whether matched datasets of transcriptomes and proteomes from bacteria under diverse conditions can be modularized in the same way to reveal novel relationships between their compositions. We find that; (1) the modules of the proteome and the transcriptome are comprised of a similar list of gene products, (2) the modules in the proteome often represent combinations of modules from the transcriptome, (3) known transcriptional and post-translational regulation is reflected in differences between two sets of modules, allowing for knowledge-mapping when interpreting module functions, and (4) through statistical modeling, absolute proteome allocation can be inferred from the transcriptome alone. Quantitative and knowledge-based relationships can thus be found at the genome-scale between the proteome and transcriptome in bacteria.

Omics data types and measurement methods emerged in the late 1990s and early 2000s. Transcriptomes were measured using hybridization to DNA arrays, and proteomes were measured using mass spectrometry. Early attempts to correlate these two omics types were unsuccessful due to complex post-transcriptional and post-translational regulation or to various technical challenges with the measurement technologies[1–3]. Later, in the mid to late 2010s, several studies compared the levels of transcripts and protein abundance on a per-gene basis[4–6]. Such correlations were achieved for a few transcript-protein pairs in humans and yeast[5,6] but proved to be more scalable in *Escherichia coli*[4]. These studies suggested that correlations between the two omics data types are possible on a small scale.

In the late 2010s, a massive number of RNAseq datasets accumulated for bacterial transcriptomes. This data deluge led to the application of machine learning methods to decompose the bacterial transcriptome into regulatory signals[7]. Of these methods, independent component analysis (ICA), a source signal extraction algorithm, was found to modularize the transcriptome into lists of independently modulated genes, termed iModulons[8]. A traditional use case of ICA is illustrated in Fig. 1A, where recording devices in a noisy room can discern the sources of noise and their contributions to the measured noise. In a biological context, an expression profile of a given sample is analogous to a microphone, since it is recording combined effects from transcriptional regulators in a noisy environment (Fig. 1B, Fig. 1C). When applied to transcriptomic data, the output of ICA was shown to most successfully match known regulons in a comparison between 42 machine learning methods[7]. Moreover,

[1]Department of Bioengineering, University of California, San Diego, La Jolla, CA 92093, USA. [2]Department of Pharmacology, University of California, San Diego, La Jolla, CA 92093, USA. [3]Skaggs School of Pharmacy and Pharmaceutical Sciences, University of California, San Diego, La Jolla, CA 92093, USA. [4]Novo Nordisk Foundation Center for Biosustainability, Technical University of Denmark, Kemitorvet, Building 220, 2800 Kgs, Lyngby, Denmark. ✉e-mail: palsson@ucsd.edu

iModulons could be integrated with known binding sites of transcriptional regulators, and compared to their associated regulons (see iModulonDB.org)[9].

iModulons from disparate datasets were shown to be similar, indicating that they represent a fundamental decomposition of the transcriptional regulatory network into underlying regulatory signals[10]. iModulons could be knowledge-enriched, thus yielding a fundamental understanding of the composition of the transcriptome and how it changes between conditions[11–17]. ICA has now been applied to several organisms across the phylogenetic tree[9]. This advance led to discoveries of gene functions[18], effects of mutations on protein complex regulation[19], and identifying energetic trade-offs across sample conditions[20]. Thus, the knowledge-based modularization of the bacterial transcriptome has led to major advances in understanding its systems characteristics.

This knowledge-based decomposition of the transcriptome naturally leads to the question: can we similarly modularize the proteome? In the present study, we generate and collect proteomic profiles for *E. coli*, modularize this dataset using ICA, and compare the iModulons in the transcriptome to those found in the proteome. This comparison leads to a large-scale, mechanistic interpretation of the relationship between the two omics data types.

## Results

### Independent component analysis modularized the proteome

We performed ICA on a compendium of proteomics samples (termed ProteomICA) consisting of 64 proteomes from a previous study[21], and 98 new samples representing conditions matching RNAseq samples in the transcriptomic compendium Precision RNA Expression Compendium for Independent Signal Extraction (PRECISE)[22]. These samples contain abundances of 1390 proteins. Since proteomic methods only capture the highest abundance proteins, this is a much lower number than the 4257 genes for which RNAseq finds transcripts[23]. The 98 new proteomic samples introduced new growth conditions representing varying stressors, carbon sources, and supplementations. These new conditions were chosen based on iModulon activities in PRECISE in order to obtain informative matched omics samples that improved signal extraction for ProteomICA (Fig. 2A). ProteomICA has 162 high-quality reproducible proteomes from *E. coli*.

ProteomICA consists of only high-quality samples with biological replicates having Pearson correlation coefficients greater than 0.90. In contrast, biological replicates in PRECISE have $R^2$ values greater than 0.95. This difference in reproducibility is in part due to the higher experimental variation in replicate proteome samples as opposed to transcriptome samples[24,25]. This difference can also be seen with the higher correlation coefficients between randomly chosen PRECISE samples than between randomly chosen ProteomICA samples (Fig. 2B, Supplementary Fig. 1). These characteristics, in turn, with higher technical noise during data generation[26], result in the ProteomICA compendium having a lower overall explained variance from the independent components and principal components than PRECISE (Fig. 2C, Supplementary Fig. 2).

The ICA decomposition of the ProteomICA database resulted in 41 proteomic iModulons (piModulons). These piModulons represent the statistically independent protein expression signals found across all 162 samples (81 unique conditions in duplicate) in the ProteomICA compendium. These piModulons represent 25% of detected proteins by count and 22% of the proteome by mass.

The 41 piModulons are classified into different categories (Fig. 2D). We find that 25 of the 41 piModulons correspond to known regulators with well-documented biological functions. Additionally, there are two piModulons that represent a specific biological function without an associated regulator. These two biological piModulons, in conjunction with the 25 regulatory piModulons, explain 46% of the overall explained variance in ProteomICA (Fig. 2D, E). Eight of the remaining 41 piModulons are considered technical and are single gene iModulons, whereas six of the remaining 41 are uncharacterized with no clear function. These final 14 piModulons represent 9% of the overall explained variance in ProteomICA. Thus, taken together, the 41 piModulons explain 55% of the variation in ProteomICA.

Since ICA is a blind source separation algorithm that deconvolutes mixed signals[27], it performs better if the signal strengths vary notably between samples[8,10]. We see a higher coefficient of variation (CV) in mass fractions of individual proteins found to be in a piModulon (Fig. 2F) versus those that are not in a piModulon (Fig. 2G). Proteins not in a piModulon account for 78% of the total proteome, with 72% being considered invariant, with CVs less than 1 ($n = 891$ proteins). In contrast, proteins in a piModulon account for 22% of the proteome with only 47% being considered invariant ($n = 163$ proteins).

Within the invariant non-piModulon proteins, we see the most abundant protein translation elongation factor, TufA[28], and outer membrane proteins OmpF and OmpA. On the other hand, MetE, coding for homocysteine transmethylase, is a very large protein that catalyzes the final step of methionine biosynthesis in the absence of cobalamin[29], is found in a piModulon due to methionine supplementation conditions that vary its activity. The overall distribution of protein mass fractions is slightly higher for piModulon proteins (median = 0.000198) than proteins not in a piModulon (median = 0.000155).

### iModulons are annotated to biologically meaningful functions

The iModulons of the transcriptome have annotated biological functions and most have transcriptional regulators associated with them (iModulonDB.org). The main method for determining the regulatory role of transcriptomic iModulons (tiModulons) is to use the corresponding established regulon in conjunction with the highly weighted genes (in a column of the matrix **M**) to see if there is a significant overlap[8,9]. The same approach was used here in the analysis of ProteomICA (Fig. 2H, I). However, due to the small number of samples in ProteomICA compared to PRECISE (162 proteomes vs 1035 transcriptomes, respectively) and fewer proteins than transcripts being identified (1390 proteins vs 4257 genes), fewer signals are decipherable from the proteomic data. As a result, some piModulons represent a combination of more than one tiModulon.

We illustrate the comparison of the two types of iModulons using two specific examples (Fig. 2H, I). The MetJ piModulon overlaps with the MetJ regulon, and the LeuO/Lrp piModulon overlaps with the Lrp or LeuO regulons. In these two examples, the LeuO/Lrp piModulon consists of the union of the Leucine and Lrp regulons, and the *metE* gene is enriched in both the MetJ and LeuO/Lrp piModulons. The corresponding columns in the iModulon matrix, **M**, contain the weightings for each gene in an iModulon.

The activities for each piModulon are found in the corresponding rows of the matrix **A**. The elements of this row can be used to plot a bar chart that shows the relative activity of the piModulon under a given condition. This bar chart is referred to as the *activity spectrum* for the piModulon (Fig. 2J, K). The activity spectrum for the MetJ piModulon shows that it exhibits low activity in samples with methionine (5 mM) supplementation and LB media, and high activity during low temperatures and adaptive laboratory evolution (ALE) under temperature stress (Temp ALE). The high activities at low temperatures are due to the first step of methionine biosynthesis, homoserine o-succinyltransferases (MetA), being more stable at lower temperatures[30]. The LeuO/Lrp piModulon also has low activity in samples with leucine (5 mM) supplementation and LB media, but also with methionine (5 mM) supplementation due to the additional *metE* enrichment. The latter's signal is not as strong due to a lower gene

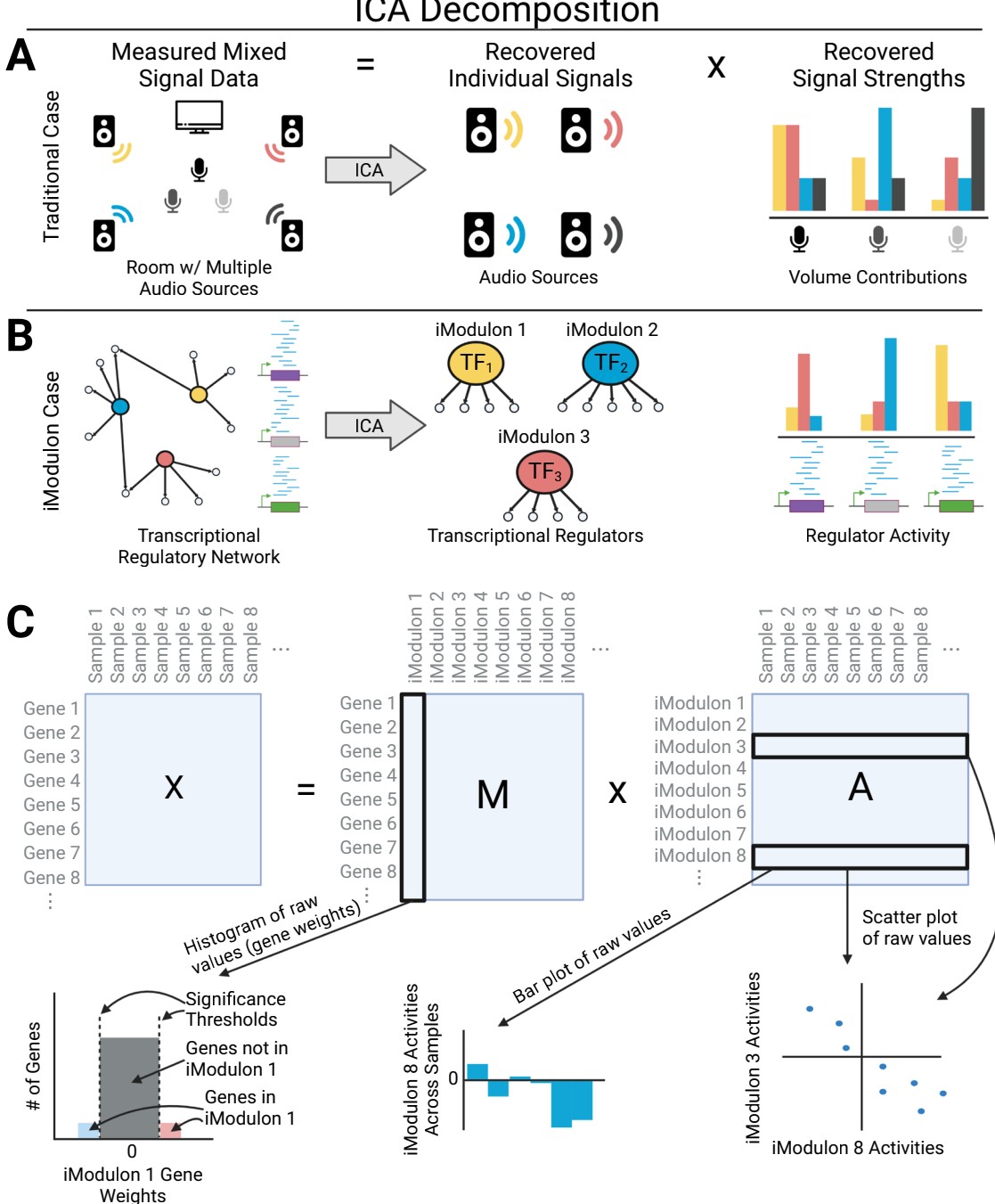

**Fig. 1 | Independent component analysis (ICA) extracts individual signals and their strengths from measured mixed signal data. A** Three microphones record the sounds in a noisy room with four audio sources. ICA is able to recover the original audio sources and their relative volume contributions to the signal captured by each microphone. **B** The transcriptional regulatory network is analogous to the noisy room, and expression data for genes is analogous to the microphones. In this case, ICA is able to recover the transcriptional regulators and their activity that contributes to each gene expression. **C** Matrix decomposition representation of the iModulon Case in panel (**B**). The matrix **X** is your measured mixed signal data (e.g., RNAseq or proteomics). The matrix **M** contains gene weights for all genes. iModulon membership is determined by thresholding the gene weights for all genes in the column. Genes that lie outside the threshold are considered in the iModulon and regulated by the transcription factor noted in panel (**B**). The matrix **A** contains all the iModulon activity values or the regulator activity level noted in panel (**B**). A row of (**A**) can be bar plotted to show the activities across samples, or it can be scatter plotted against another row of (**A**) to compare iModulon activities.

weight of *metE* in the piModulon, and thus, does not see as negative an activity compared to the leucine and LB samples.

Thus, ProteomICA can be decomposed into piModulons using ICA. If there is a corresponding compendium of matched transcriptomic samples available, then the iModulons computed from both can be related to one another (see the following section).

## iModulons in the proteome mirror those in the transcriptome
The correlation between iModulon gene weights enables the comparison of the weighted gene content and allows us to match corresponding piModulons and tiModulons computed from PRECISE and ProteomICA (Fig. 3A). We computed Pearson correlation coefficients between gene weights for all pairs of piModulons and tiModulons. pi- and tiModulons are only considered matched if the resulting

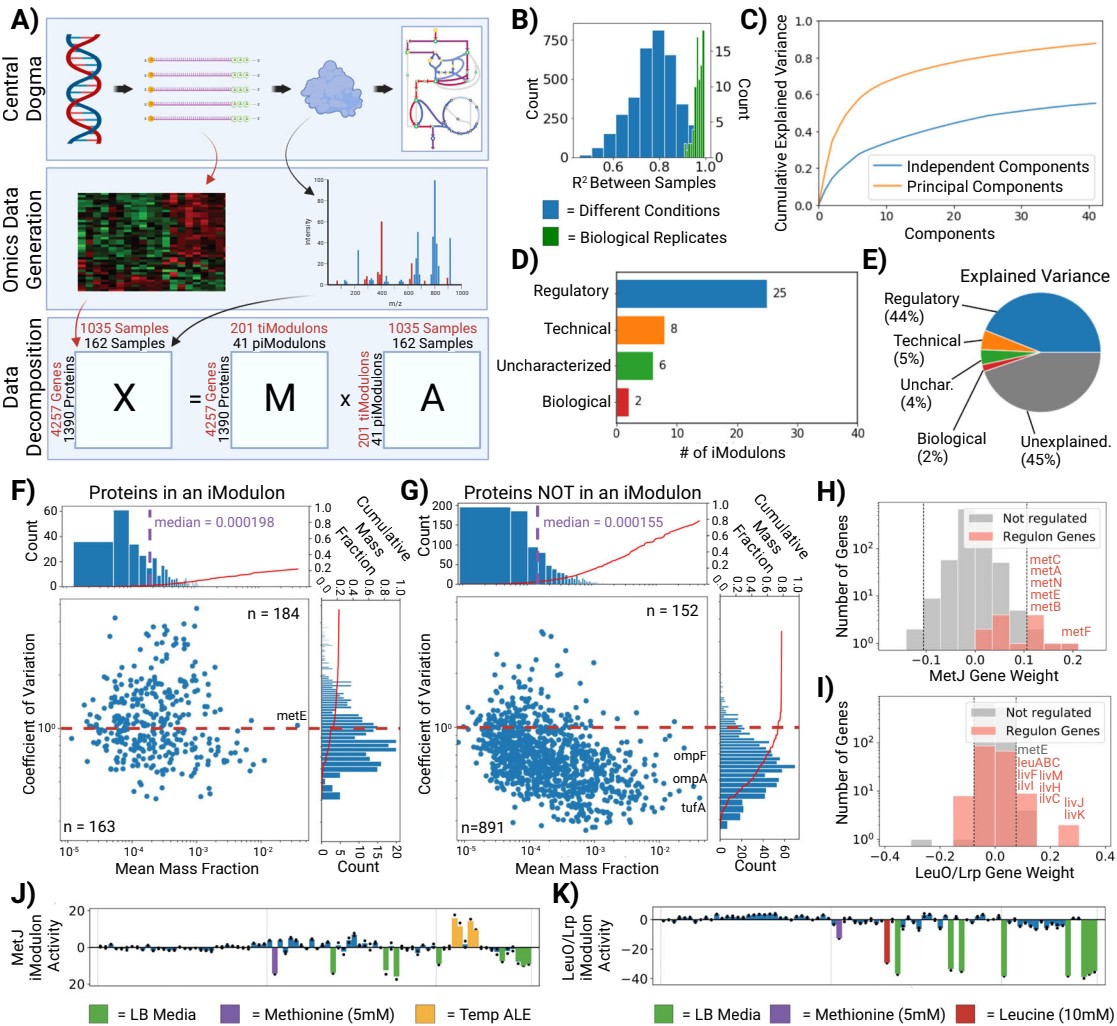

**Fig. 2 | ICA of a compendium of proteomic samples (ProteomICA). A** The central dogma of molecular biology is the process whereby genetic information is converted into functional proteins that catalyze metabolic reactions and carry out other cellular functions. Genome-wide datasets can be generated for the transcriptome and proteome and analyzed using ICA. Given a matrix of gene expression or of protein abundance, **X**, ICA identifies independently modulated groups of genes or proteins called iModulons (expressed as weights contained in a column of **M**). Every sample in the dataset has an activity associated with each iModulon that becomes condition-specific (row of **A**). Matrix multiplication of **M***A results in **X**. **B** Histogram of Pearson correlations between proteomic samples (biological replicates vs. random samples). **C** Cumulative explained variance of the independent components and principal components from matrix decomposition of the proteomics compendium. **D** Enrichment categories for proteomic iModulons. **E** Pie chart of the explained variances for each enrichment category. **F**, **G** Scatter plots of the coefficient of variation (CV) and mean mass fractions for proteins in an iModulon, and NOT in an iModulon, respectively. Axes graphs show a histogram of the distribution and the cumulative mass fraction of proteins (red). The dashed red line indicates a CV of 1. Proteins below this threshold are considered invariant. N counts for each section are listed. **H**, **I** Histogram of the gene weights within the MetJ and LeuO/Lrp independent components (column of **M**), respectively. The significance threshold (gray) identifies the most extreme values, and thus which genes are considered enriched in an iModulon. Regulon genes associated with the iModulon regulator are highlighted. **J**, **K** proteomic iModulon activity spectrums for MetJ and LeuO/Lrp, respectively. Differentially activated samples are highlighted with sample condition metadata. LB: Lysogeny Broth, Temp: Temperature, Unchar: Uncharacterized. Source data are provided as the Source Data file.

correlations are above a threshold of 0.25 (Supplementary Table 1). This value is set low due to the non-uniformity of the gene-weight distributions for iModulons with similar functions across organisms or omics data types. Matches were also manually checked. A total of 17 of the 25 regulatory piModulons match with a tiModulon, in addition to both biological piModulons have a matching tiModulon. As mentioned before, some piModulons are combinations of more than one tiModulon. For most of these cases, the piModulon matched with every tiModulon within the combination. For example, the FliA/FlhDC piModulon matched with the FliA and FlhDC-2 tiModulons.

Upon sorting all iModulons within each compendium by their explained variance and comparing the genes, it becomes evident that there is a very strong correspondence between the iModulon's explained variance between the two omics data types (Fig. 3B). Of the

20 total matches between the datasets, 10 piModulons match to 15 tiModulons that are ranked in the top 25 for both, out of 41 total piModulons and 201 total tiModulons. The 10 piModulons explain 32% of the variability in ProteomICA, while the 15 matched tiModulons explain 26% of the variability in PRECISE. These values are quite similar even with the significant differences in the total number of iModulons obtained from each omic data type. Additionally, the explained variability captured by the two compendia (ProteomICA and PRECISE) is 55% vs 83%, respectively. Thus, the piModulons detect the stronger signals, but cannot detect the more silent signals that the tiModulons can.

Matched iModulon pairs can be described based on the recall they have of each other's gene sets (Fig. 3C). Recall for each matched pair of iModulon can be calculated using the ratio of the number of genes in

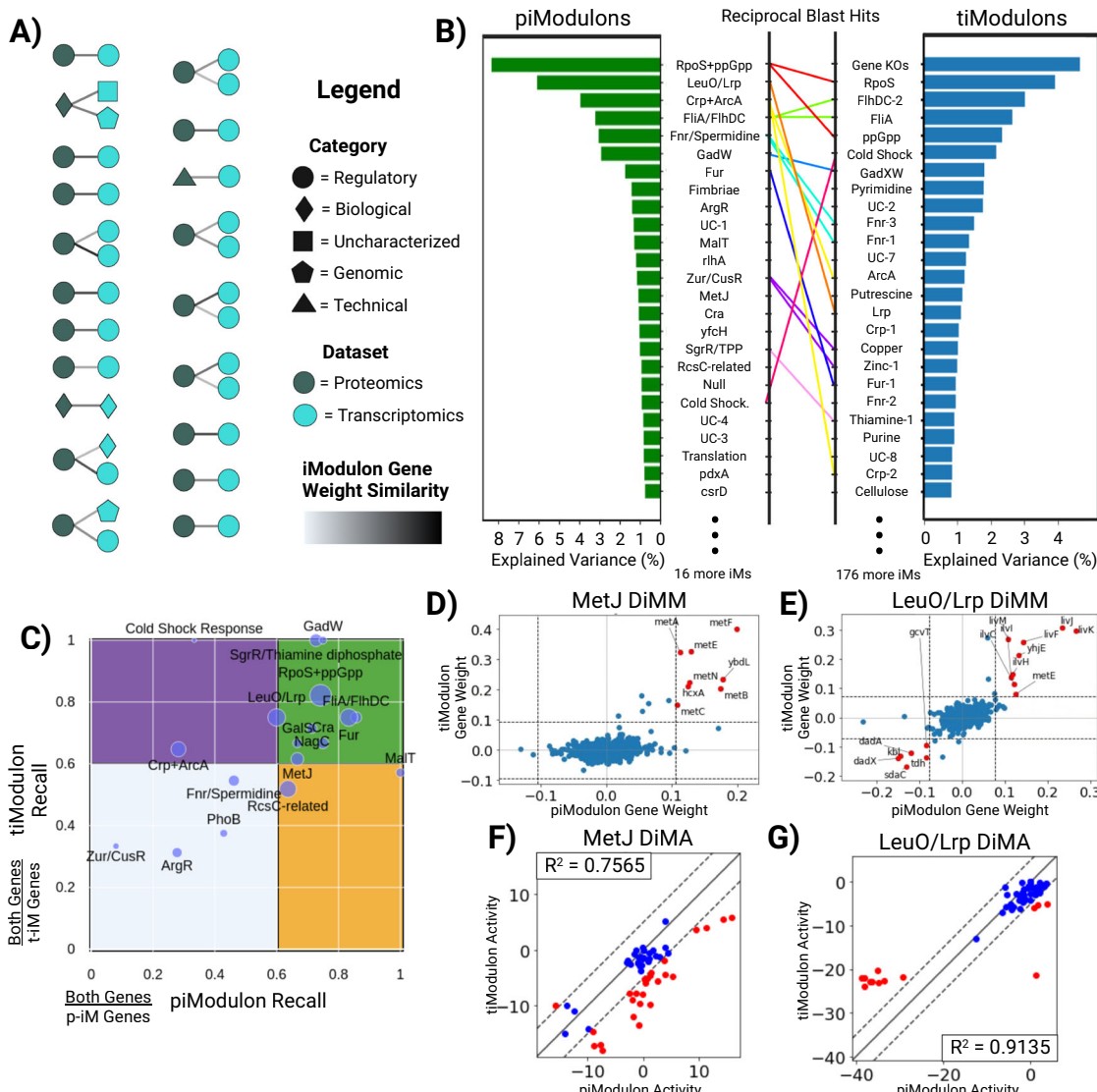

**Fig. 3 | Proteomic iModulons (piModulons) exhibit similar gene lists and activity levels as transcriptomic iModulons (tiModulons). A** A schematic of correlations between the independent components within the transcriptomic compendium (PRECISE) and the proteomics compendium (ProteomICA). Colors indicate the dataset, while the shapes indicate the enrichment category. The shade of the links is determined by the similarity of the two independent components (Pearson correlation) that match between the two datasets. **B** Ranked bar plots of the explained variances for each iModulon within both datasets. Matches between the top-ranked iModulons are shown with rainbow connections. Lumped piMo-dulons that match to multiple tiModulons have multiple connections of the same color. **C** Scatter plot of the piModulon recall and tiModulon recall for all matches between the two datasets. 'tiModulon Genes' are genes enriched in the transcriptomic iModulon. 'piModulon Genes' are genes enriched in the proteomic iModulon. 'Both Genes' is the intersection of the tiModulon and piModulon genes. The size of the point is determined by the number of 'Both Genes' **D**, **E** Differential iModulon Membership (DiMM) plots that compare the gene weights for the matched piModulons and tiModulons for MetJ and LeuO/Lrp, respectively. The significance threshold (gray) shows which genes are enriched in each iModulon. Red genes are enriched in both iModulons. **F**, **G** Differential iModulon Activity (DiMA) plots that compare the activities for the matched piModulon and tiModulons for MetJ and LeuO/Lrp, respectively. Activities are considered differentially activated for samples that lie outside the significance threshold (gray). Differentially activated samples are highlighted in red. Source data are provided as the Source Data file.

both iModulons ('Both Genes') to the number of genes in the piMo-dulon or tiModulon (piModulon Recall and tiModulon Recall, respectively). Larger iModulons mostly fall in the high recall green quadrant, whereas smaller iModulons predominantly fall in the light blue low recall quadrant (Fig. 3C). Regulon recall for tiModulons with larger regulons is typically poor[12,13], but that is not observed here, indicating strong correspondence between matched iModulons.

The iModulon matrix **M** and the activity matrix **A** can also be compared for each matched pairs of iModulons. A differential iModulon membership plot (DiMM) compares the gene weights (column of **M**) for matched iModulons between PRECISE and Proteo-mICA (Fig. 3D, E). Genes enriched in both iModulons are highlighted in red, and ICA is able to identify the same genes in both compendia regardless of gene weight sign. A differential iModulon activity plot (DiMA) compares the activities for condition-matched samples in PRECISE and ProteomICA (Fig. 3F, G). Correlations between the two activities are calculated with differentially activated samples highlighted in red. Correlations range from strong to weak depending on the number of differentially activated samples and are explored more in the following section.

We thus find that there is good correspondence between matched piModulons and tiModulons, with the former often representing combinations of the latter. The gene composition of matched pi- and tiModulons is congruent, and so are their condition-dependent

activities. This correspondence of modularization of the transcriptome and proteome enables deeper analysis.

## Matched iModulons reflect established regulatory mechanisms

The ti- and piModulons can be compared in terms of the gene weights (i.e., composition of the signal) and their activity levels (i.e., signal strengths), see Fig. 3D–G. Plotting the differential iModulon activities (DiMA plots) of all pairs of matched ti- and piModulons reveals three distinct groups; pairs that are (1) transcriptome-dominant (signal more active in the tiModulon than the piModulon), (2) proteome-dominant, and (3) neutral. These differences can be interpreted in light of known transcriptional and translational regulation that, in some cases, is condition-specific, but in many cases are broad and well established. All DiMA plots can be found in Supplementary Fig. 3. We describe a few cases in detail. The following reviews also provide full descriptions for each type of regulatory mechanism in case readers may not be familiar with them[31–33].

Higher tiModulon activities indicate transcriptional attenuation, riboswitches, or transcript stability that lead to relatively higher RNA than protein: When tiModulon activities are higher than that of the matched piModulon, the iModulon has a stronger signal in the transcriptome and is said to be transcriptome-dominant. This characteristic can be attributed to transcriptional attenuation, riboswitches that inhibit translation, or stability due to structural reorganization.

LeuO/Lrp (Fig. 4A): The LeuO/Lrp iModulon contains the *leuLABCD* operon, which consists of leucine synthesis genes. The operon is known to be regulated by ribosome-mediated attenuation in the presence of charged leucine tRNAs[34]. Thus, we observe this mechanism as a transcriptome-dominant iModulon: in rich media or leucine-supplemented minimal media, the expressed RNA is not translated, leading to an upregulation of the tiModulon relative to the piModulon. We also observed the iModulon becoming proteome-dominant in the case of arginine supplementation, which may indicate competitive repression by arginine of the dual regulator Lrp and its associated operons.

SgrR/Thiamine diphosphate (Fig. 4B): Thiamine diphosphate (TPP) can act as a ligand that binds to a riboswitch that inhibits the translation of the *thiMD* and *thiCEFSGH* operons[35,36]. Most samples in rich LB media are differentially active towards their respective tiModulon activities, probably due to thiamine-induced premature transcriptional termination. Additionally, samples grown on galactose, pyruvate, and fumarate have higher overall pi- and tiModulon activities due to increased demand for the thiamine cofactor in essential reactions pyruvate dehydrogenase complex and 2-oxoglutarate (2-ketoglutarate) dehydrogenase complex.

Cold Shock Response (Fig. 4C): At temperatures below 37 °C, the *cspA* mRNA undergoes temperature-dependent structural reorganization[37]. This structural change is likely due to the stabilization of an otherwise thermodynamically unstable folding intermediate. At low temperatures, the structure is also less susceptible to degradation[38]. Samples at 30 °C are differentially active and transcriptome-dominant, while samples at 42 °C have no activity due to mRNA instability at high temperatures.

Higher piModulon activities indicate translation activation, protein product autoregulation, or riboswitches that lead to relatively higher protein than RNA: When piModulon activities are higher than that of the matched tiModulon, the iModulon has a stronger signal in the proteome and is said to be proteome-dominant. This characteristic can be attributed to riboswitches that promote translation, protein products autoregulating transcription, or transcript inhibition due to other proteins.

RpoS+ppGpp (Fig. 4D): RpoS, the major stress-related sigma factor, and guanosine 3,5-bispyrophosphate (ppGpp), an important alarmone, both act as master regulators for a wide range of genes including those involved in oxidative stress, temperature shock, acid

stress, starvation, and osmotic stress[39]. Both regulators integrate several stress signals, and ppGpp helps stabilize RpoS, leading to complex transcriptional regulation[39]. In addition, ppGpp was recently found to directly activate the translation of some genes[40]. Thus, we observe a proteome-dominant expression pattern for this iModulon in several samples. Samples from ALE are also known to have low RpoS activities, which is replicated here with both pi- and tiModulon activities[8].

MetJ (Fig. 4E): MetJ regulates methionine synthesis genes at the transcriptional level in response to methionine and related molecules[41]. As expected, both the tiModulon and the piModulon are therefore downregulated in LB media and with methionine supplementation. One member of this iModulon, MetA (homoserine o-succinyltransferase, the first step of methionine biosynthesis), is inherently unstable under stressful conditions and high temperatures[30]. Thus, it is regulated by temperature-dependent proteolysis[42]. Interestingly, conditions which make the protein more stable, such as low temperatures and heat-tolerant strains with *metA* mutations (Temp ALE), are proteome-dominant for this iModulon. This observation likely reflects the increased stability of the MetJ-regulated proteins in those conditions.

Fnr/Spermidine (Fig. 4F): Spermidine is a known small molecule that binds to a riboswitch that facilitates the translation of the *oppABCDF* operon[43]. Like the other polyamines, putrescine, and spermine, it stimulates the assembly of 30S ribosomal subunits, increasing general protein synthesis up to 2-fold[44]. Here, we see more samples with higher piModulon activities than tiModulon activities due to this increase.

ti- and piModulons that contain their regulator show similar signal strengths in the transcriptome or proteome: When tiModulon activities are similar to that of the matched piModulon, the ti- and piModulons have similar strengths in the transcriptome and proteome and is said to be neutral. This can be illustrated by looking at the Cra, NagC, GalS iModulons (Fig. 4G–I). The DNA-binding transcriptional dual regulator Cra is a member of both the Cra ti- and piModulon. The GlcNAc tiModulon contains its transcriptional regulator NagC, as well as the Galactose tiModulon which contains GalS. It's an uncommon phenomena for an iModulon to contain its own regulator because regulators typically don't exhibit linear relationships with the genes they regulate due to the complexity of the TRN. If an iModulon exhibits unique characteristics, such as temporal dynamics, the regulatory function is split into multiple iModulons in which case one will contain the regulator and the others won't (e.g., Phosphate-1,2; FlhDC-1,2,3; NtrC-1,2,3; Fnr-1,2,3)[9,22]. When the regulator is in an iModulon of a single function that didn't split, it indicates that there aren't any complex regulatory interactions since the iModulon activity is correlated with its regulator's expression. In the case of Cra, NagC, and GalS, this leads to an overall neutral relationship between the proteome and transcriptome as seen here.

Previous studies have clearly shown that tiModulons can be knowledge-enriched by mapping known transcription factor binding sites in promoters of genes found in a tiModulon[9]. The results presented in this section take knowledge enrichment a step further. Namely, various molecular mechanisms are reflected in the relative activity levels of pi- and tiModulons. Thus, the ability of ICA to detect these regulatory mechanisms enables us to knowledge-enrich the relationships between iModulons. When piModulon and tiModulon activities are not well-correlated, they indicate post-transcriptional regulatory events. Many such events have been previously characterized in the literature, as described in this section.

## tiModulon activities allow prediction of proteome allocation

Revealing the relationships between tiModulons and piModulons opens up the possibility of predicting the composition of the proteome straight from RNAseq data. Such predictions would be advantageous since the composition of the transcriptome can be measured

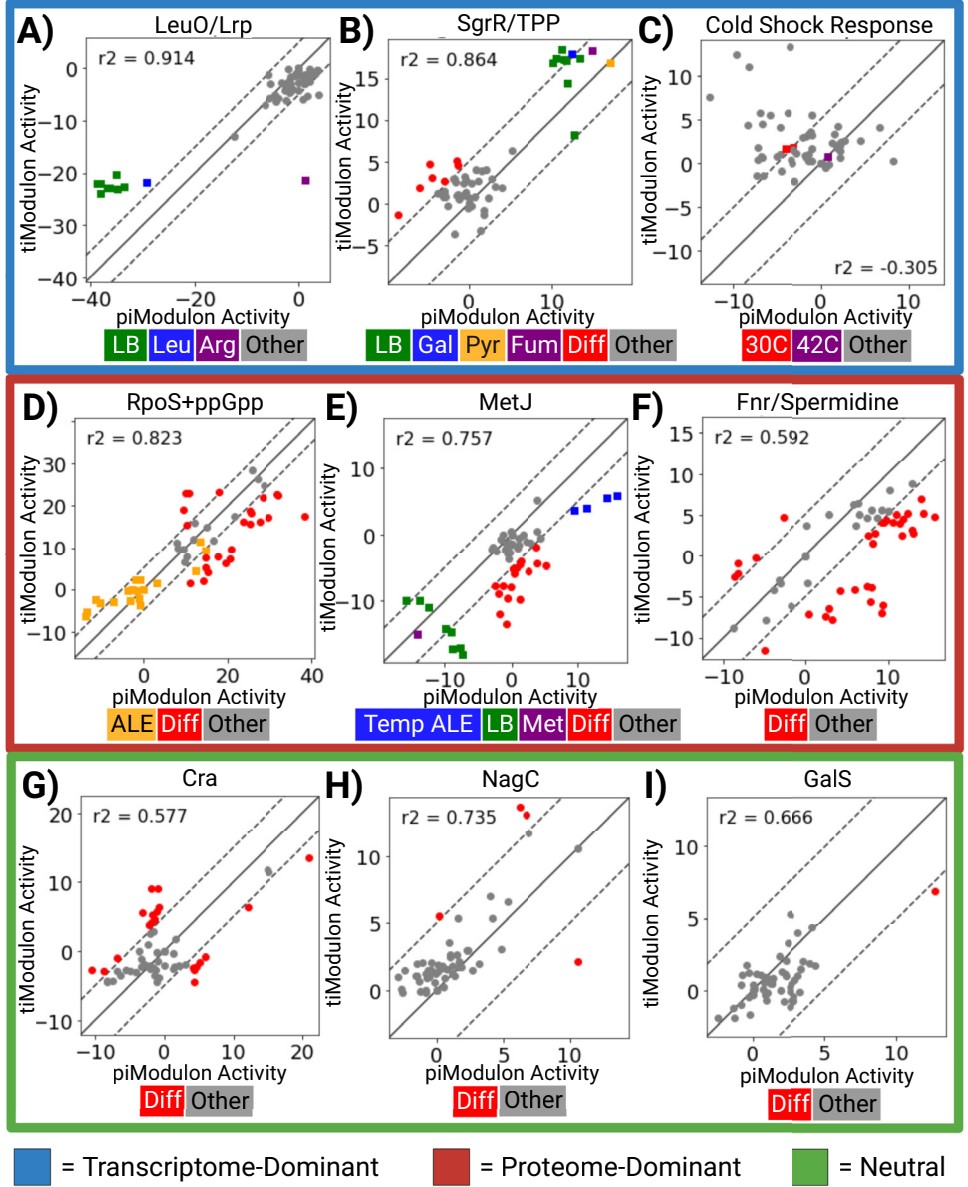

**Fig. 4 | Comparing piModulon and tiModulon activities for matched samples reveal condition-specific regulatory mechanisms.** Differential iModulon Activity (DiMA) plots for some matched iModulons between the two datasets. Plots are categorized by the observed result due to regulatory mechanisms, such as riboswitches, transcriptional attenuation, temperature-dependent transcript structural reorganization, and protein product autoregulation. **A–C** iModulons that are transcriptome-dominant (signal more active in the tiModulon than the piModulon) are highlighted in blue. **D–F** iModulons that are proteome-dominant are highlighted in red. **G–I** iModulons that are neutral are highlighted in green. Activities are considered differentially activated for samples that lie outside the significance threshold (dashed line). Legends for each plot are placed below each plot. LB: Lysogeny Broth, Leu: Leucine Supplement, Arg: Arginine Supplement, Gal: Galactose Carbon Source, Pyr: Pyruvate Carbon Source, Fum: Fumerate Carbon Source, Diff: Differentially Activated, Temp: Temperature, Met: Methionine Supplement. $n = 57$ conditions. Source data are provided as the Source Data file.

cheaper, faster, and with higher precision and accuracy than the composition of the proteome.

We thus sought to find quantitative relationships between RNA-seq data and proteome allocation. Three types of relationships (linear, exponential, broken line, Fig. 5A) were identified by plotting tiModulon activities against the mass fraction of the proteome allocated to the genes represented by the tiModulon. tiModulon activities that are linearly correlated with their proteome allocation indicate proteome-optimized sectors (i.e., amino acid biosynthesis). tiModulon activities that are exponentially correlated with their proteome allocation indicate proteome-optimized, yet expensive sectors (i.e., stress-related responses). Finally, tiModulon activities that fit a broken line indicate a

thresholding response due to phenomena like bet-hedging (e.g., central carbon metabolism)[45–50].

tiModulon activities that represent strong correlations with their proteome allocation account for 565 genes, 243 of which are covered by ProteomICA (Fig. 5B). tiModulon activities are considered to have a strong correlation with proteome allocation if the adjusted $R^2$ of the regression fitting after leave-one-out cross-validation is above 0.3 (Fig. 5C). The adjusted $R^2$ 0.3 cutoff was selected as a corrected threshold to use between the three fitting types, as it is equivalent to an $R^2$ of 0.7 for linear regression in this dataset. These 243 genes account for, on average, 26% of the proteome in the compendia, with a CV of 0.19 (Fig. 5D). All of the scatter plots and regressions for tiModulons

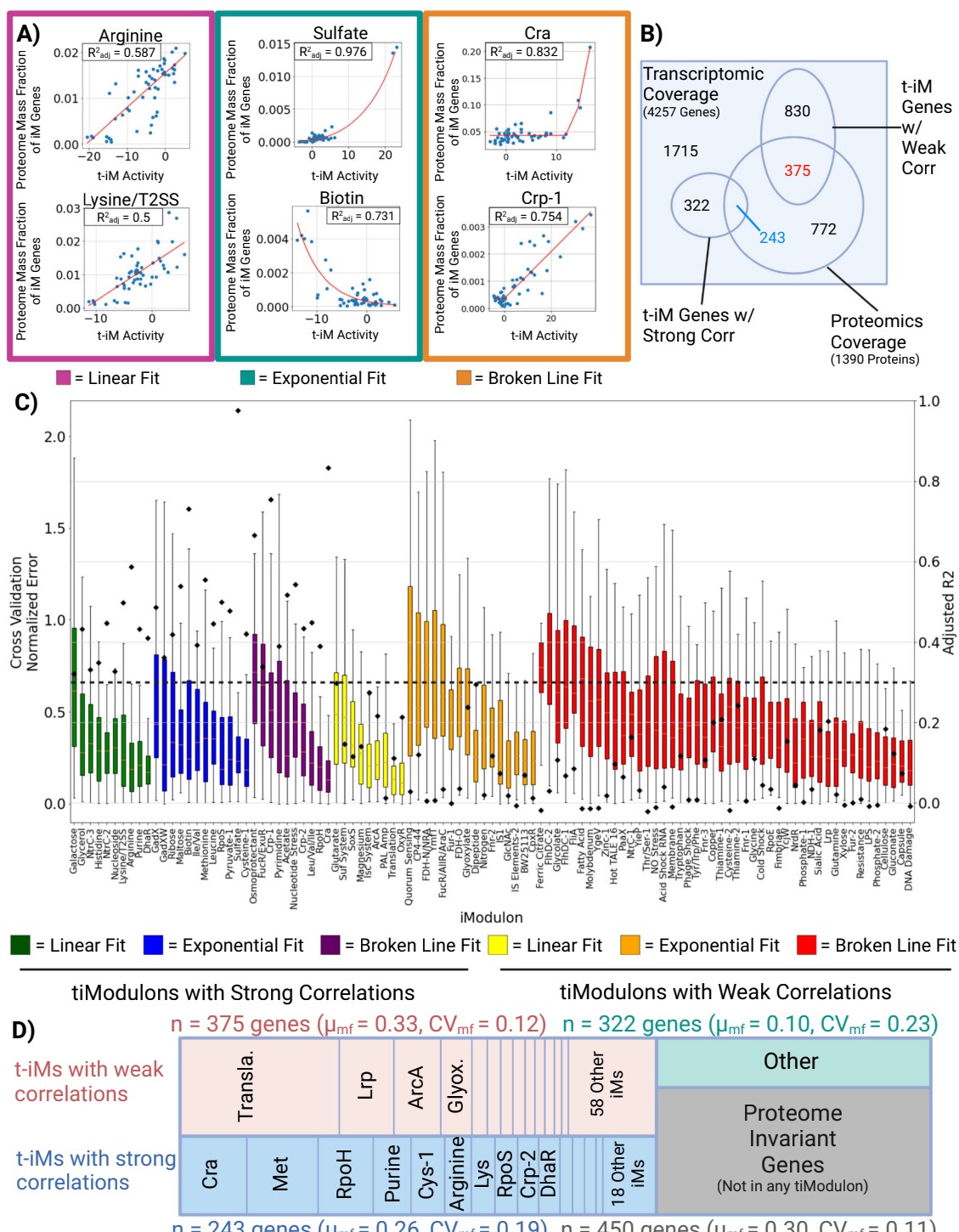

**Fig. 5 | Predictability of proteome allocation using tiModulon activities.**
**A** Scatter plots for selected tiModulon activities and their measured proteome mass fraction of the associated enriched genes. tiModulons are characterized based on which regression method resulted in the best adjusted $R^2$ value. The three fits were linear, exponential, and broken line. **B** Venn diagram detailing the number of genes/proteins covered by both datasets, in addition to the regression results. A tiModulon is considered to have a strong correlation with its proteome allocated if the adjusted $R^2$ value ≥ 0.3. tiModulon genes covered by proteomics data with strong correlations are in blue, while covered tiModulon genes with weak correlations are in red. **C** Boxplots showing the distribution of normalized errors after

cross-validation $\left(\frac{|ypred-ytest|}{yavg}\right)$. Final model adjusted $R^2$ values are scattered on top of the boxplots with black diamonds. iModulons are organized/colored by their fitting method and quality of the fitting. $n = 57$ conditions for each model. Boxplot minima, maxima, medians, and percentiles can be found in Source Data. **D** Treemap of the proteome allocation using the tiModulon regression results. tiModulons with strong correlations are in blue, while those with weak correlations are in red. Genes that are not in a tiModulon and whose protein has a CV ≤ 1, are invariant and labeled in gray. Genes that are not in any tiModulon nor proteome invariant are labeled in green. Source data are provided as the Source Data file.

with strong correlations can be found in Supplementary Fig. 4. tiModulon activities that represent weak correlations with their proteome allocation account for 1205 genes, 375 of which are covered by ProteomICA (Fig. 5B). These genes account for, on average, 33% of the proteome in the compendia, with a CV of 0.12 (Fig. 5D). Proteome invariant genes that are not in a tiModulon account for 30% of the proteome (CV = 0.11), and genes that are not in a tiModulon but are not invariant account for 10% of the proteome (CV = 0.23).

We also wanted to ensure that these results and methods were robust, so we did additional analysis on our regression methods with holdout splitting percentages ranging from 10% to 30% in 5% increments on top of leave-one-out cross-validation (Supplementary Fig. 5, Supplementary Fig. 6).

Being able to infer absolute proteome allocation from the transcriptome alone, regardless of condition, requires generalizable statistical models with large adjusted $R^2$ values. Normalized cross-validation error for each regression is not statistically significant between strongly and weakly correlated tiModulons, yet the adjusted $R^2$ values differ quite substantially due to outliers in both datasets that cause large errors. While these weaker regressions cannot be used for generalization, unlike the stronger regressions, some can still be used to estimate the proteome allocated for the specific conditions that do not fall in outlier conditions. For example, the translation tiModulon has one of the weakest correlations but accounts for 11% of the proteome, but due to a small number of outliers is categorized as weak (Supplementary Fig. 7). Removal of the outlier conditions would categorize the tiModulon as strong and enable inference of proteome allocation.

Taken together, these results show that ICA decomposition of the transcriptome enables inference for 56% of the proteome allocation for general cases (gray and blue sectors, Fig. 5D), with up to an additional 33% being inferrable for specific conditions (red sector, Fig. 5D). These relationships include the effects of post-transcriptional regulation and should represent practical ways of estimating how differential regulation of gene expression affects the proteome composition. These results provide a strong impetus for generating larger proteomic datasets to generate stronger and broader correlations between the datasets and proteome allocation.

## Discussion

Recent advances in big data analytics have enabled the knowledge-enriched modularization of the transcriptome for various microbial species[8,11–17]. Here we investigated if matched datasets of transcriptomes and proteomes could be modularized in the same way to reveal novel relationships between their compositions. Using ICA analysis of matched datasets we found that; (1) the modules of the proteome and the transcriptome are comprised of a similar list of gene products, (2) the modules in the proteome often represent combinations of modules from the transcriptome, (3) known transcriptional and post-translational regulation is reflected in differences between two sets of modules, allowing for knowledge-mapping when interpreting module functions, and (4) through statistical modeling, absolute proteome allocation can be inferred from the transcriptome alone.

Modularizing the proteome via ICA decomposition has resulted in biologically meaningful groups of independently modulated genes, termed proteomic iModulons, or piModulons. They are similar to previous studies that have successfully modularized the transcriptome for various organisms using ICA[9]. We show that the piModulons have a similar gene composition as transcriptomic iModulons or tiModulons. While the proteomics compendium used, ProteomICA is newer and has five times fewer samples than the transcriptomics compendium used, PRECISE[22], the former produces detectable signals in just under five times the number of independent modules computed from the latter. This result suggests that if we expand and improve the quality of

ProteomICA, perhaps with the inclusion of additional post-translational modifications in search parameters, we may be able to achieve a higher fidelity understanding of the regulation of proteome allocation.

Due to this size limitation, a number of identified piModulons from ProteomICA represent combinations of tiModulons. While this may seem problematic at first, a similar phenomenon is visible when decomposing the transcriptome at lower dimensionalities[51], and it has been shown that iModulons tend to split as more conditions are added, enabling ICA to identify more signals in the datasets[8]. It is quite promising to see that a number of these combined modules in the proteome represent the highest explained variance in the compendia of both data types. PRECISE has a total of 201 tiModulons that explain 83% of the variance in the dataset, while ProteomICA has a total of 41 piModulons that explain 55% of the variance in the dataset, but the top-ranked iModulons that are matched between both represent 26% and 32% of the variance, respectively. Note that ICA-derived explanations are based on knowledge, or mechanisms, in contrast to the explanation of statistical variation that is obtained using principal component analysis (PCA).

The congruence of the gene compositions of matched ti- and piModulons led to the comparisons of their activity levels. Such comparison enabled further knowledge enrichment of the matched sets of iModulons over and above their individual annotation with regulatory knowledge. The comparison allowed the attribution of a number of established transcriptional and post-translational regulatory mechanisms. Regulatory phenomena, such as riboswitches and attenuation, are easily identifiable when comparing matched iModulons of the corresponding regulatory component. Thus, interoperable data analytics at the genome-scale can capture an increasing number of established regulatory mechanisms through detailed molecular biology studies.

We also showed that it is possible to utilize transcriptomic datasets to infer proteome allocation. Previously, this was only achievable on a per-gene basis, but modularization via ICA has scaled up the scope to sets of genes enabling inference of proteome re-allocation[4–6]. Transcriptomic samples can be measured cheaper, faster, and with higher precision and accuracy than proteomics samples. The fact that we can demonstrate a correlation between tiModulon activity levels and proteome allocation with this method in its infancy, provides a strong impetus to explore how broadly we can achieve this correlation which requires the generation of larger matched sets of transcriptomic and proteomic datasets. As we generate more matched sets, our regression models will become increasingly more robust since we only currently have 57 matched conditions for regression. In its current form, we train the regression models on all data points using leave-one-out cross-validation due to our dataset size limitation. However, the inclusion of additional holdouts (Supplementary Fig. 5, Supplementary Fig. 6) doesn't significantly deteriorate the proteome allocation prediction capacity and in fact, further supports the robustness of our initial models. More data will enable more rigorous validation testing of these regression models in follow-up studies.

Furthermore, enabling the prediction of proteome re-allocation between conditions using transcriptomics can further bridge the gap between observable physiological states and molecular profiling methods. Genome-scale computational models that compute proteome allocation can now be parameterized better and thus be used to build quantitative relationships between the regulation of gene expression and physiological functions and fitness[52,53].

Taken together, we have shown that ICA can modularize the transcriptome and proteome in a consistent manner. The iModulons are knowledge-enriched and thus interpretable based on the fundamentals of cell and molecular biology. This achievement enables the meaningful interoperability of two key omics data types, leading to quantitative and knowledge-based relationships at the genome-scale

between the proteome and transcriptome. This capability, in turn, gives us a deep understanding of the systems biology of bacteria, which leads to interpreting their adaptation and changes to environmental stimuli. Thus, distal and proximal causation can be studied at a new scale to more deeply understand organism fitness and survival strategies.

## Methods

### Proteomic sample preparation

Frozen cell pellets were resuspended in lysis buffer (75 mM NaCl (Sigma–Aldrich), 3% sodium dodecyl sulfate (Fisher Scientific), 1 mM sodium fluoride (VWR International, LLC), 1 mM β-glycerophosphate (Sigma–Aldrich), 1 mM sodium orthovanadate, 10 mM sodium pyrophosphate (VWR International, LLC), 1 mM phenylmethylsulphonyl fluoride (Fisher Scientific), 50 mM HEPES (Fisher Scientific) pH 8.5, and 1× complete EDTA-free protease inhibitor mixture). Samples were vortexed and sonicated (Qsonica, Q500 equipped with a 1.6-mm microtip) at 20% amplitude for three cycles of 2 s of sonication followed by 2 s of rest, with a total sonication time of 12 s.

Total protein abundance was determined using a bicinchoninic acid Protein Assay Kit (Pierce) as recommended by the manufacturer. Six micrograms of protein were aliquoted for each sample. Sample volume was brought up to 20 μL in a solution of 4 M Urea and 50 mM HEPES, pH = 8.5. Proteins were reduced and alkylated with 5 mM dithiothreitol (DTT) for 30 minutes at 56 °C and 15 mM iodoacetamide (IAA) at room temperature in the dark for 20 min. The reaction was quenched with the addition of 5 mM DTT for 15 min at room temperature in the dark. Proteins were precipitated by adding 5 uL of 100% trichloroacetic acid on ice for 10 min, then centrifuged at 16,000 × g for 5 min at 4 °C. The supernatant was removed, and pellets were washed gently in 50 uL of ice-cold acetone. The wash was repeated twice, and the pellets were dried on a heating block at 56 °C. Pellets were resuspended in 1 M Urea and 50 mM HEPES, pH 8.5. The UPS2 Standard (Sigma) was reconstituted as follows: 20 μL of 4 M Urea and 50 mM HEPES, pH 8.5 was added to the stock tube and vortexed and sonicated for 5 min each. Reduction and alkylation were performed as described above. The standard was then diluted in 50 mM HEPES, pH 8.5 such that the final concentration of urea was 1 M. Then 0.88 μg of the standard was spiked into each experimental sample. Samples were then digested first with 6.6 μg of LysC at room temperature overnight followed with 1.65 ug sequencing grade trypsin (Promega) for 6 hours at 47 °C. Digestion was terminated with the addition of 3.3 μL 10% trifluoroacetic acid (TFA) and was brought to a final volume of 300 uL with 0.1% TFA. Samples were centrifuged at 16,000 × g for 5 min and desalted with in-house-packed Stage-Tips[21,54]. Samples were then dried in a speedvac, and stored at −80 °C until LC–MS/MS.

### LC–MS/MS

Samples were resuspended to 1 μg/μL in 5% acetonitrile (ACN) and 5% formic acid (FA), vortexed, and sonicated. Samples were analyzed on an Orbitrap Fusion Mass Spectrometer with in-line Easy NanoLC (Thermo) in technical triplicate. Samples were run on an increasing gradient from 6 to 25% ACN + 0.125% FA for 75 min, then 100% ACN + 0.125% FA for 10 min. One microgram of each sample was loaded onto a 35 cm length in-house–pulled and –packed glass capillary column (ID 100 μm, OD 360 μm) heated to 60 °C. The column was triple packed first with C4 resin (5 μm, 0.5 cm, Sepax), then C18 resin (3 μm, 0.5 cm, Sepax), and finally C18 resin (1.8 μm, 29 cm, Sepax). Electrospray ionization was achieved through application of 2000 V to a stainless-steel T-junction connecting the sample, waste, and column. The mass spectrometer was run in positive polarity mode with MS1 scans performed in the orbitrap (375 m/z to 1500 m/z, 120,000 resolution, AGC set to $5 \times 10^5$, ion injection time of 100 ms maximum, dynamic exclusion set to 30 s duration). Top N was used for fragment ion isolation, with N set to 10. A decision tree was used to isolate ions with a charge

state of two between 375 m/z and 1500 m/z, and ions with charge states of 3–6 were isolated between 600 m/z and 1500 m/z. Precursor ions were fragmented using fixed collision-induced dissociation and fragment ions were detected in the linear ion trap in profile mode. Target AGC was set to $1 \times 10^4$.

Technical triplicate spectral data was searched against custom reference proteomes of the respective strains (see above) with the UPS2 database appended using Proteome Discoverer 2.5 (Thermo). Spectral matching and an in-silico decoy database were performed using the SEQUEST algorithm[55]. Precursor ion mass tolerance was set to 50 PPM, and fragment ion tolerance was set to 0.6 Daltons. Trypsin and LysC were specified as digesting enzymes with a maximum missed cleavage of two sites allowed. Peptide length was set between 6 and 144 amino acids. Dynamic modifications included the oxidation of methionine (+15.995 Da), and static modifications included carbamidomethylation of cysteines (+57.021 Da). A false discovery rate of 1% was applied during spectral searches.

### Proteome abundance estimations

The protein abundance estimation steps used on the new dataset are the same used on the previous PXD015344[21]. The top3 metrics were calculated for each protein as the average of the three highest peptide areas[5,56]. Linear regression was used to calibrate the top3 metric with the UPS2 standard according to the following model:

$$\log_{10}(A) = a + b \log_{10}(top3)$$

Where A is the amount of loaded protein A and top3 is the average of the three highest peptide areas. In order to obtain abundance relative to cell dry weight, we used the following formula:

$$C_i = \gamma \frac{A_i}{\sum_j A_j}$$

Where the numerator of the ratio, $A_i$, is the abundance of the ith protein, and the denominator is the sum of abundances for all j proteins. We use a constant ratio $\gamma = 13.94$ umol*gDW$^{-1}$ [57].

### Compiling ProteomICA and data imputation

Upon estimating protein abundances, proteins with <50% coverage within the dataset were removed. Of the remaining proteins, samples with no abundances were replaced with the minimum global protein abundance. Datasets were then converted to mass fractions or protein concentrations and concatenated to compile the proteomics compendia. Similar to how the final transcriptomics expression compendium is log-transformed $\log_2(TPM + 1)$, the final proteomics expression compendium is scaled by a million and also log-transformed similarly $\log_2(PPM + 1)$[8]. Biological replicates with $R^2 < 0.9$ were removed to reduce technical noise. Individual datasets were then centered using a common reference condition between all datasets to reduce batch effects.

### Independent component analysis

ICA was run following the PyModulon workflow. ICA is implemented using the optICA extension of the popular algorithm FastICA. The script can be found (https://github.com/SBRG/iModulonMiner/tree/main/4_optICA). The output of the algorithm are two matrices, **M** and **A**, given an input matrix **X**. In our case, the matrix **X** is our curated proteomics compendium. The matrix **M** contains robust independent components, and the matrix **A** contains their corresponding activities.

### Computing independent components and their enrichments

After running ICA and obtaining the resulting matrix decomposition, the PRECISE1K workflow (https://github.com/SBRG/precise1k)[22] is used to choose the optimal dimensionality of the resulting ICA runs

and associate regulator enrichments to iModulons. After automation, enriched iModulons are checked for their associated regulators and uncharacterized iModulons are manually curated using a variety of annotation tools such as COG and GO terms.

## Comparing iModulons between PRECISE and ProteomICA

The PyModulon python package (https://github.com/SBRG/pymodulon)[8] was used for the DiMM, and DiMA, and explained variance plotting functions. The PyModulon package also enables comparison between organisms via the compare_ica function, which was utilized to correlate the iModulons between PRECISE and ProteomICA. The compare_ica function uses only the overlap of the two gene sets to calculate correlations, and has a default threshold of 0.25 which was not changed. The ICA workflow was run on a subset of PRECISE that contained only the 1390 genes covered by proteomics and 137 matched samples (57 conditions without replicates), centered using the same common reference condition as ProteomICA. Population samples were not included in the matched dataset, only clones. The resulting decomposition was used for DiMM, DiMA, and recall plots.

## Regression and cross-validation for proteome allocation

Only matched samples that are present in both PRECISE and ProteomICA were used for this analysis. Uncharacterized, Genomic, and Technical tiModulons were ignored. For each tiModulon, tiModulon activity was plotted against its associated proteome mass fraction. Replicates were averaged for both iModulon activity and proteome allocation. Leave-one-out cross-validation was performed with three different fits, linear, exponential, and broken line. The model with the lowest mean average error was selected as the best fitting method for that tiModulon. To analyze the robustness of the models and results, data was split into train and test groups, with the test group ranging from 10% to 30% in increments of 5%. Leave-one-out cross-validation was conducted on the training set, with the final model parameters evaluated against the test set.

## Calculating proteome allocated to groups of tiModulons

Proteome allocation to groups of tiModulons was calculated sequentially to avoid multiple iModulon gene memberships. First, tiModulons with strong correlations were sorted in descending order by model performance. tiModulon gene lists were extracted and used to calculate the mean and CV for mass fractions across the compendia. After which, the genes were removed from the total gene list (1390 genes in the proteome coverage) and could no longer be used for another tiModulon. tiModulons with weak correlations were calculated in a similar fashion after iterating through all the tiModulons with strong correlations. Lastly, proteome invariant genes and then 'Other' were calculated straight from the remaining gene list since there could no longer be overlap.

## Statistics and reproducibility

No statistical method was used to predetermine size. Two samples were excluded from analysis due to not meeting the biological replicated threshold (see Compiling ProteomICA and data imputation section). The experiments were not randomized. The Investigators were not blinded to allocation during experiments and outcome assessment.

## Reporting summary

Further information on research design is available in the Nature Portfolio Reporting Summary linked to this article.

## Data availability

All MS-based proteomics raw files for newly run samples are available on the ProteomeXchange Consortium with the dataset identifier PXD039558. All previously used proteomics data can be found under the identifier PXD015344. Their protein mass fractions are available in Supplementary Data 1. ProteomICA decomposition matrices, iModulon, and sample table are available in Supplementary Data 2. PRECISE1k subset decomposition matrices are available in Supplementary Data 3. A matched sample table is available in Supplementary Data 4. The full PRECISE1k decomposition matrices are available at https://github.com/SBRG/precise1k. Raw RNA-seq data used in PRECISE1k have been deposited at GEO and are publicly available. Accession numbers are listed in the metadata file located in the GitHub repository at the path: data/precise1k/metadata_qc.csv. Source data are provided with this paper.

## Code availability

Code for our Independent Component Analysis pipeline can be found on GitHub (https://github.com/SBRG/iModulonMiner, https://github.com/SBRG/precise1k). Code for all iModulon analysis can also be found on GitHub (https://github.com/SBRG/pymodulon).

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

## Acknowledgements

The work was funded by the Novo Nordisk Foundation Grant Number NNF20CC0035580, the National Institute of General Medical Sciences of the National Institutes of Health Grant R01 GM057089, and by the generous support of the Y.C. Fung Endowed Chair. We would like to thank Daniel Zielinski and Cameron Lamoureux for their helpful discussions. We would also like to thank Marc Abrams for their help with manuscript proofreading. Figures were created using Biorender.com.

## Author contributions

A.P., A.V.S., and B.O.P. designed the study. D.M., Y.H., A.C., D.J.G., and S.M. performed experiments. A.P. analyzed the data. A.P., K.R., and B.O.P. wrote the manuscript with contributions from all other co-authors.

## Competing interests

The authors declare no competing interests.
