## [Peer Review File · Nature Communications]

Proteome allocation is linked to transcriptional regulation through a modularized transcriptomeReviewers' Comments:

Reviewer #1:

Remarks to the Author:

The manuscript from Patel et al. describes an extension of the iModulon framework to proteomics data, previously confined to transcriptome data. This framework uses independent component analysis (ICA), a statistical method that separates a multivariate signal, here sets of protein abundances, into their underlying independent sources, here representing modulons or regulons (here referred to as iModulons).

Basically, the analysis is a decomposition of the matrix of abundances of *Escherichia coli* proteins samples under diverse growth conditions (X) into two matrices, representing the iModulons these proteins cluster into (M) and the activity of the single proteins for the tested conditions (A).

They nicely show that ICA identified iModulons from the proteome data that largely overlap with those from transcriptome data highlighting a strong relationship between transcripts and proteins. While some data show linear correlation as one would infer from the central dogma of molecular biology, as has previously been shown, many proteins show a differential activity from the transcript counterpart. The authors identify several modulons enriched with differentially activated proteins and explain these with (known) regulatory mechanisms.

The authors further aimed to delineate a quantitative relationship between transcript activities and proteome allocation. While the attempt is promising it is currently suffering from low coverage and smaller proteome data sets compared to available RNAseq data.

Overall, the authors present an exciting extension of the iModulon with promising prospects for utilising omics data to enrich knowledge of regulatory programs in cells and further improve the predictability of metabolic models. However, there are some shortcomings that should be addressed by the authors to improve comprehensibility and readability of the work, as stated below.

Comments

Page and line numbers are missing which makes it hard to refer to the sections I am commenting on.

1. Providing a brief description of the Independent Component Analysis (ICA) analysis and the interpretation of the resulting matrices would be particularly beneficial for readers who are not familiar with the iModulon framework or the associated publications. This would enhance the accessibility of the study's findings and help readers to better understand how the data were analyzed and interpreted without looking up the technical details in the Method section or worse checking additional literature.

2. Section "Matched iModulons reflect established regulatory mechanisms"

This section contains some inaccuracies in the language used and would benefit from substantial improvement. Although the term "expression" is often used interchangeably to refer to both transcription and translation, it would be clearer and more comprehensible to use "gene expression" or simply "transcription" to refer to the process of RNA synthesis, and "protein synthesis" or "protein production" to refer to the process of translation. Additionally, it appears that "transcription" may have been mistakenly used in place of "translation" at times, and it is important to ensure that the correct terminology is used consistently throughout the section. A thorough review of the text is necessary to ensure the accuracy and clarity of the language used.

p. 10/11 in the PDF:

A clearer description of the regulatory mechanisms discussed in this section is necessary for readers to fully understand the study's findings. The use of illustrations or diagrams could also help to support the text and make the regulatory mechanisms more easily understandable. This would be particularly helpful for readers who may not be familiar with the specific regulatory processes being discussed.

LeuO/Lrp:

- It is unclear why transcriptional attenuation would result in a transcriptome-dominant iModulon, as

the generated transcripts would be truncated and therefore not contribute to the overall transcriptome. It would be helpful to provide a more detailed explanation of how the data were analyzed to support this finding. This would help readers to better understand the relationship between transcription attenuation and the observed iModulon.

- Wrong termini used? RNA is translated not transcribed
- Unclear which transcriptional regulators (activators or repressors) are competitively repressed by arginine

SgrR/Thiamine diphosphate:

- In cases, thiamine seems to be wrongly used for thiamine diphosphate.
- Thiamine diphosphate as well as spermidine (p. 11) are not riboswitches, which are RNA-based regulatory devices but can be ligands to a riboswitch and alter its conformation.

Ti- and piModulons with similar strength:

- A more detailed explanation is needed to clarify why modulons that contain their regulator are uncommon and why these are often split into sub-modulons.

2. The inference of protein (re-)allocation from transcriptome data is an intriguing approach, and the data presented in this study is promising. However, it is important to note that the method is still in its infancy and has significant limitations, which should be emphasized more clearly. Also, the authors are asked to state, which dataset (size, quality – both were described to have limitations) are required.

Supplementary:

There is a difference between the size of the matrices both for proteome and transcriptome data and number of modulons and samples reported in the main text. This needs clarification/ reconciliation.

Additional comments

- Some gene names not in italics
- Discussion, 2nd paragraph: there are ~5-times more tiModulons than piModulons. Correct.
- Discussion: Discourage to speak of first principles (but rather fundamental).
- Methods: Sometimes, methods are described as optional ("can be used") which creates ambiguity about what has been done here. Clearly state how the study has been performed.

Reviewer #2:

Remarks to the Author:

In this work, Patel et al. report their efforts to relate transcriptome and proteome in *E. coli*. This is a work from Palsson's lab and they build upon previous works that applied ICA to transcriptomic data by now applying it to proteomic data. This allows them to develop a framework to relate transcriptome and proteome based on the ICA decomposition of each dataset. The work is interesting as provides new insights into how the modules of the transcriptome relate with those of the proteome and enable the inference of the proteome allocation from the transcriptome.

However, the paper is very complex and obscure in some parts, mainly for readers not familiar with the previous work of this group. These uninitiated readers will have the feeling that most of the workflow and methods are black boxes and will find difficulties following the whole process. A figure describing the whole workflow is important to guide the reader. Improving the methods description and general writing will help the reader too. A brief introduction to key concepts and methods reported in previous works is important too.

A major problem is an ample difference in sample sizes between proteomICA and PRECISE (162 proteomes vs 1035 transcriptomes) which the authors recognize as problematic as it causes some piModulons to represent a combination of more than one tiModulon. While the authors discuss that this could be due to post-transcriptional regulation, it is not clear whether it is the case or this is just an artifact of the proteome's low resolution.

Specific concerns:

- Page 2: How the conditions of the new 98 proteomic samples were "matched" with the RNAseq conditions? What were the "matching" criteria?
- Page 3: How the 41 piModulons were classified (Figure 1D)?
- Page 5: How the piModulons activities were computed from matrix A? The activities are the raw value present in each cell or was it processed in some way?
- Page 5 and Figure 2A: I do not know what I am seeing. I understand this figure shows the relations among both datasets and their annotated categories, but why nodes are not labeled? The relations showed are all the possible ones? Why some relations are repeated?
- Page 5: Pearson correlations were computed between piModulons and tiModulons, but the number of genes in each iModulon is different, and the same size vectors are required to compute Pearson. How did you solve this? Did you only correlate the genes in the overlap of pi- and ti-Modulons? How the 0.25 threshold was established?
- Page 6 and Figures 2D and 2E: What is the rationale behind the 0.1 thresholds chosen?
- Figure 4A: Why does the exponential fit for Sulfate activity clearly shows a couple of elbows? It is misleading.
- Figure 4C: The caption misses mentioning what the diamonds mean. I assume it is the R^2_{adj} , right? Given that the number of tiModulons having weak correlations with broken line fits is larger than the others, I suggest separating strong and weak correlations using lines just below the x-axis, and not just depending on the colors as it could be confusing.
- Page 14 and Figure 4C: a line showing the baseline at 0.3 could improve the figure.
- Page 15: Regarding the translation tiModulon, which are the outlier conditions? What is the relation between them and the non-outliers?
- Page 18 'Compiling ProteomICA and data imputation': How many biological replicates were removed? What was the common reference condition used?

Reviewer #3:

Remarks to the Author:

This work from Patel et al. applies Independent Component Analysis (ICA) on both proteome and transcriptional datasets to find independent modules (41 modules and 201 are found, respectively), to identify 25 pairs with a Pearson Correlation Coefficient (PCC) larger than 0.25. Then, the authors use curve-fitting to elucidate putative regulatory mechanisms.

The paper is well-written, and effort has been put to create quality figures and accompanying materials. The topic is interesting.

Major Issues:

1. There is some ambiguity and limited justification on validity of the modules. Their discovery is based on arbitrary thresholds (0.25 or $R^2 > 0.3$), while systematic benchmarking and validation of the resulting modules are missing. In addition, the relationship between the sample size and the number of discovered independent modules should be provided.
2. From all E. coli genes, only 25 pairs are found that satisfy a PCC threshold of 0.25 (which is low to begin with). It is not clear why PCC and ICA would be the right methods to be applied in this problem, to begin with.
3. The significance of the results (that are clearly stated in the abstract) is not clear. So modules in proteome/transcriptome have common genes, and modules at one layer are combination of another, which is to be expected and I am not sure what the lesson here is. The last result that "absolute proteome allocation can be inferred from the transcriptome alone", would be very impactful and amazing but I don't see support in this paper for accurate inference of proteome from transcriptome, piece-wise or otherwise.

4. Statistical support is generally limited throughout the claims - e.g. for interpretation of DIMA plots, or putative regulatory mechanisms based on matched modules.

5. The paper uses different regression models to showcase the predictability of proteome allocation using ti-Modulon activities. The authors provide results using leave-one-out-cross-validation while also making sure that replicates of the same samples are not in the various folds. It is not clear what is the size of the validation set and whether a holdout was used for model selection.

Minor comments:

1. The authors distinguish genes as having a strong correlation if the corresponding regression models have $R^2 > 0.3$. Again the justification of that threshold is missing. The authors state that strong-correlation genes significantly differ from weak-correlation genes, but no p-value is given. Therefore, it is not clear what is the essential difference between strong and weak correlation genes and their impact on regression models. It will be interesting to analyze what is the biological reasons for their distinguishment.

2. Although the PRECISE datasets (with identified 92 independent modules) are thoroughly verified and published, much more modules (201 independent modules) are identified in the transcriptome dataset (PRECISE1K) used in this study and the explanation of the different amount of identified independent modules is limited. In addition, the PRECISE1K dataset is not peer-reviewed.

3. There are some inconsistencies (iModulon and i-Modulon) and I found it difficult to follow the decomposition results of the subset (Supplementary Data 3) as there is no explanation of what the resulting 60 components are and the relationship between these components and 41 components extracted from the entire dataset.

4. It would be great to provide observations regarding matching between ti- and pi- modules, or the curve fitting results which are in contrast to existing knowledge.

Reviewer #4:

None

Reviewer #5:

None

Reviewers' Comments: **Black Text**

Author's Planned Response: **Blue Text**

Reviewer #1 (Remarks to the Author):

The manuscript from Patel et al. describes an extension of the iModulon framework to proteomics data, previously confined to transcriptome data. This framework uses independent component analysis (ICA), a statistical method that separates a multivariate signal, here ses of protein abundances, into their underlying independent sources, here representing modulons or regulons (here referred to as iModulons).

Basically, the analysis is a decompositions of the matrix of abundances of Escherichia coli proteins samples under diverse growth conditions (X) into two matrices, representing the iModulons these proteins cluster into (M) and the activity of the single proteins for the tested conditions (A).

They nicely show that ICA identified iModulons from the proteome data that largely overlap with those from transcriptome data highlighting a strong relationship between transcripts and proteins. While some data show lienar correlation as one would infer from the central dogma of molecular biology, as has previously been shown, many proteins show a differential activit from the transcript counterpart. The authors identify several modulons enriched with differnetially activated proteins and explain these with (known) regulatory mechanisms.

The authors further aimed to delineate a quantitative relationship between transcript activities and proteome allocation. While the attempt is promising it is currently suffering from low coverage and smaller proteome data sets compared to avaiabale RNAseq data.

Overall, the authors present an exciting extension of the iModulon with promising prospects for utilising omics data to enrich knowledge of regulatory programs in cells and further improve the predictability of metabolic models. However, there are some shortcomings that should be addressed by the authors to improve comprehensibility and readability of the work, as stated below.

We thank the reviewer for their time and greatly appreciate their feedback. The summary you provided captures the paper very well. We hope that the changes made to the manuscript make it easier to read and comprehend.

Comments

Page and line numbers are missing which makes it hard to refer to the sections I am commenting on.

Page and line numbers have been added to the revision.

1. Providing a brief description of the Independent Component Analysis (ICA) analysis and the interpretation of the resulting matrices would be particularly beneficial for readers who are not familiar with the iModulon framework or the associated publications. This would enhance the accessibility of the study's findings and help readers to better understand how the data were analyzed and interpreted without looking up the technical details in the Method section or worse checking additional literature.

Many thanks for the suggestion. We have added another figure to the introduction which explains the basics of the iModulon framework and provides some more context for how the ICA outputs are used in this manuscript.

2. Section "Matched iModulons reflect established regulatory mechanisms"
This section contains some inaccuracies in the language used and would benefit from substantial improvement. Although the term "expression" is often used interchangeably to refer to both transcription and translation, it would be clearer and more comprehensible to use "gene expression" or simply "transcription" to refer to the process of RNA synthesis, and "protein synthesis" or "protein production" to refer to the process of translation. Additionally, it appears that "transcription" may have been mistakenly used in place of "translation" at times, and it is important to ensure that the correct terminology is used consistently throughout the section. A thorough review of the text is necessary to ensure the accuracy and clarity of the language used.

We thank the reviewer for identifying these issues and bringing them to our attention. We have gone through this section and ensured that "expression" is not used vaguely. Additionally, we thoroughly reviewed the text and corrected any unintentional swapped transcription/translations.

p. 10/11 in the PDF:

A clearer description of the regulatory mechanisms discussed in this section is necessary for readers to fully understand the study's findings. The use of illustrations or diagrams could also help to support the text and make the regulatory mechanisms more easily understandable. This would be particularly helpful for readers who may not be familiar with the specific regulatory processes being discussed.

We appreciate the author's suggestion. Due to the space limitations of the figure and manuscript, we opted to instead provide reviews to the reader that describe each type of mechanism for readers not familiar with the specifics. A sentence was added in the first paragraph of the section to let readers know.

LeuO/Lrp:

- It is unclear why transcriptional attenuation would result in a transcriptome-dominant iModulon, as the generated transcripts would be truncated and therefore not contribute to the overall transcriptome. It would be helpful to provide a more detailed explanation of how the data were analyzed to support this finding. This would help readers to better understand the relationship between transcription attenuation and the observed iModulon.

We thank the reviewer for bringing this point up. While the transcripts would be truncated, they can still affect the quantification of the transcriptome due to fragmentation bias of partial transcripts. The samples that we believe are affected by transcriptional attenuation have very low iModulon activities (<-20) indicating that there is essentially no activity at the RNA or protein level anyway.

- Wrong termini used? RNA is translated not transcribed

We have changed 'transcribed' to 'translated'

- Unclear which transcriptional regulators (activators or repressors) are competitively repressed by arginine

We have edited the main text to make this clear

SgrR/Thiamine diphosphate:

- In cases, thiamine seems to be wrongly used for thiamine diphosphate.
- Thiamine diphosphate as well as spermidine (p. 11) are not riboswitches, which are RNA-based regulatory devices but can be ligands to a riboswitch and alter its conformation.

We appreciate the reviewer's clarifications and have edited the text to fix any inaccuracies.

Ti- and piModlons with similar strength:

- A more detailed explanation is needed to clarify why modulons that contain their regulator are uncommon and why these are often split into sub-modulons.

The main text has been edited to clarify this.

2. The inference of protein (re-)allocation from transcriptome data is an intriguing approach, and the data presented in this study is promising. However, it is important to note that the method is still in its infancy and has significant limitations,

which should be emphasized more clearly. Also, the authors are asked to state, which dataset (size, quality – both were described to have limitations) are required.

We have edited the main text in the discussion to more clearly emphasize what the reviewer has mentioned.

Supplementary:

There is a difference between the size of the matrices both for proteome and transcriptome data and number of modulons and samples reported in the main text. This needs clarification/ reconciliation.

We greatly appreciate the reviewer taking the time to go through the supplemental files.

DatasetS2 contains the ProteomICA matrices. These matrices have 162 samples, 41 iModulons, and 1390 proteins. These numbers are reflected in the main text.

DatasetS3 contains the PRECISE1k subset matrices, which is not the same as PRECISE1k that is described in the main text (1035 samples, 201 iModulons, and 4257 genes). This decomposition is described in the Methods section “Comparing iModulons between PRECISE and ProteomICA”. The full PRECISE1k matrices that you are referring to can be found at <https://github.com/SBRG/precise1k>. To avoid confusion, we have added this link to the Data Availability section.

Additional comments

- Some gene names not in italics

All gene names are now italicized and protein names are now capitalized.

- Discussion, 2nd paragraph: there are ~5-times more tiModulons than piModulons. Correct.

Yes, that is correct. The reason is that the trans

- Discussion: Discourage to speak of first principles (but rather fundamental).

We appreciate the correction and have made the change in the Discussion.

- Methods: Sometimes, methods are described as optional (“can be used”) which creates ambiguity about what has been done here. Clearly state how the study has been performed.

We have edited the language in the Methods section to be clear and unambiguous.

Reviewer #2 (Remarks to the Author):

In this work, Patel et al. report their efforts to relate transcriptome and proteome in *E. coli*. This is a work from Pálsson's lab and they build upon previous works that applied ICA to transcriptomic data by now applying it to proteomic data. This allows them to develop a framework to relate transcriptome and proteome based on the ICA decomposition of each dataset. The work is interesting as provides new insights into how the modules of the transcriptome relate with those of the proteome and enable the inference of the proteome allocation from the transcriptome.

However, the paper is very complex and obscure in some parts, mainly for readers not familiar with the previous work of this group. These uninitiated readers will have the feeling that most of the workflow and methods are black boxes and will find difficulties following the whole process. A figure describing the whole workflow is important to guide the reader. Improving the methods description and general writing will help the reader too. A brief introduction to key concepts and methods reported in previous works is important too.

A major problem is an ample difference in sample sizes between proteomICA and PRECISE (162 proteomes vs 1035 transcriptomes) which the authors recognize as problematic as it causes some piModulons to represent a combination of more than one tiModulon. While the authors discuss that this could be due to post-transcriptional regulation, it is not clear whether it is the case or this is just an artifact of the proteome's low resolution.

We thank the reviewer for their time and constructive comments. We agree that the content is complex, especially for those not familiar with the work of our group. A new figure has been added to the beginning of the main text which provides, what we hope, is a clearer description of the iModulon framework. Additionally, we briefly describe the basics of the ICA decomposition to help those uninitiated readers.

With respect to your last comment, we are excited by the results in this manuscript and don't believe the relationships we see are due to artifacts of low resolution or size. Regarding the size of the dataset, as mentioned in the discussion section, ICA of the *E. coli* PRECISE compendia at lower dimensionalities (akin to fewer samples) results in similar combinations of modules. Additionally, some of our smaller organisms on imodulondb.org with 100-200 transcriptomic samples also result in a similar number of identified modules as from the 162 samples from ProteomICA. With regards to the resolution of the dataset, higher coverage and thus higher resolution of our proteomics data should only improve the scope at which we identify unique relationships such as post-transcriptional regulation since more gene/protein relationships are analyzed. The exciting and informative results from the analysis of the 162 samples provide a strong impetus for generating a larger proteomic dataset that matches the 1035 transcriptomes.

Specific concerns:

- Page 2: How the conditions of the new 98 proteomic samples were “matched” with the RNAseq conditions? What were the “matching” criteria?

We chose 98 samples from PRECISE that represented conditions that we didn't have proteomics for at the time. These 98 samples were chosen because they also activated the most iModulons in PRECISE that weren't in our preliminary ProteomICA dataset. These were mostly growth conditions with stressors (ROS, Temperature), new carbon sources, and supplementations.

- Page 3: How the 41 piModulons were classified (Figure 1D)?

The 41 piModulons and their classifications can be found in Supplemental File 'DatasetS2' > Sheet 'iModulon Table' > column 'enrichment_category'

- Page 5: How the piModulons activities were computed from matrix A? The activities are the raw value present in each cell or was it processed in some way?

The values are raw from the matrix A. Neither M or A are processed. We added this clarification to the new Figure 1.

- Page 5 and Figure 2A: I do not know what I am seeing. I understand this figure shows the relations among both datasets and their annotated categories, but why nodes are not labeled? The relations showed are all the possible ones? Why some relations are repeated?

You are correct, you're seeing all possible relationships that are also detailed in the Supplementary Table 1. The nodes aren't labeled due to readability and available space in the figure. There are no repeated relationships, just some that look very similar.

- Page 5: Pearson correlations were computed between piModulons and tiModulons, but the number of genes in each iModulon is different, and the same size vectors are required to compute Pearson. How did you solve this? Did you only correlate the genes in the overlap of pi- and ti-Modulons? How the 0.25 threshold was established?

You are also correct here. Only the union of the two gene sets was used for correlations (1390 gene/proteins). The Methods section 'Comparing iModulons between PRECISE and ProteomICA' has been edited to clarify your questions. The 0.25 threshold is the default threshold value used in the function. While this may seem like a very low value, this is due to the non-uniformity of the gene-weight distributions for iModulons with similar functions across organisms or omics

datatypes. For example, you can see two direct examples with Figure 3D and Figure 3E. The correlation for LeuO/Lrp is 0.65 while MetJ is 0.44, but it's very clear based on the gene enrichments (Figure 2H and 2I) that these iModulons are related to Methionine and Leucine. Another example is Arginine and ArgR that have a correlation of 0.33. The default threshold was identified and set after analyzing other species on iModulondb.org as part of our ongoing Modulome project that aims to analyze all publicly available high-quality transcriptomics datasets using ICA.

- Page 6 and Figures 2D and 2E: What is the rationale behind the 0.1 thresholds chosen?

The thresholds are automatically calculated based on an outlier test that relies on the D'Agostino test statistic. This is automatically built into the ICA workflow described in Methods section "Computing robust Independent components and their enrichments".

- Figure 4A: Why does the exponential fit for Sulfate activity clearly shows a couple of elbows? It is misleading.

We thank the reviewer for noticing this error. The plot has been updated.

- Figure 4C: The caption misses mentioning what the diamonds mean. I assume it is the R^2_{adj} , right? Given that the number of tiModulons having weak correlations with broken line fits is larger than the others, I suggest separating strong and weak correlations using lines just below the x-axis, and not just depending on the colors as it could be confusing.

You are correct. Thank you for mentioning it, we have added it to the caption. Also, thank you for the suggestion. We tried a few iterations of what you suggested but it turned out to be too cluttered.

- Page 14 and Figure 4C: a line showing the baseline at 0.3 could improve the figure.

We thank the reviewer for the suggestion and have added it to the figure.

- Page 15: Regarding the translation tiModulon, which are the outlier conditions? What is the relation between them and the non-outliers?

The outlier conditions are evolved samples that acquired mutations, shifting them across the Translation iModulon landscape. As of right now, it's hard to determine which mutations are causing the outliers but with more data it should be clearer.

- Page 18 'Compiling Proteomics and data imputation': How many biological replicates were removed? What was the common reference condition used?

A total of 8 replicates were removed from 170 total. The common reference condition was Wild Type *E. coli* grown on glucose-minimal M9 media.

Reviewer #3 (Remarks to the Author):

This work from Patel et al. applies Independent Component Analysis (ICA) on both proteome and transcriptional datasets to find independent modules (41 modules and 201 are found, respectively), to identify 25 pairs with a Pearson Correlation Coefficient (PCC) larger than 0.25. Then, the authors use curve-fitting to elucidate putative regulatory mechanisms.

The paper is well-written, and effort has been put to create quality figures and accompanying materials. The topic is interesting.

We thank the reviewer for their time and feedback. We are glad that you think the paper is interesting and well-written. Hopefully, the changes and responses we made will address your concerns about the validity of the presented results.

As stated in the abstract, the paper has 4 basic results: 1) iModulons in the transcriptome and proteome have similar functions and gene content, 2) the proteomic modules are sometimes combinations of transcriptomics modules, 3) differences in these modules (e.g. activities) reflect known regulatory mechanisms between the two layers, and 4) proteome allocation can be inferred from transcriptomic iModulon activities. Most of the concerns the reviewer has are due to statistical power and are focused on the last result, the attempt to predict the proteome allocation from the transcriptome. We agree that there currently are limitations and the method is in its infancy. In our study, we had a limited number of matched proteomics samples. Even with a modest statistical strength, the fact that we can get a correlation between the proteome and transcriptome provides a strong impetus to generate more matched datasets. While we would have loved to have additional samples to obtain stronger correlations, as the reviewer would like to see, the generation of hundreds of more proteomes will take some time and resources and provides a strong reason to carry out a follow-up study to improve these statistics. It took 2 years to identify which samples to generate matched data for and to generate them for this manuscript. As of initial submission, our proteomic dataset was the 70th largest on MassIVE out of 14,000, and the largest transcriptomic-matched dataset.

We would like to note that the correlations between the transcriptome and proteome allocation are statistical but not mechanistic. However, underlying mechanisms are reflected in the different weightings of proteins and transcripts in the regulatory signals.

Major Issues:

1. There is some ambiguity and limited justification on validity of the modules. Their discovery is based on arbitrary thresholds (0.25 or $R^2 > 0.3$), while systematic benchmarking and validation of the resulting modules are missing. In addition, the

relationship between the sample size and the number of discovered independent modules should be provided.

We appreciate the reviewer's attention to the validity of the discovered modules. It is important to note that the discovery of the modules is not based on any correlation thresholding. The modules are automatically identified via ICA decomposition. The characterization of the modules also isn't based on any correlation thresholding. Most characterized modules are automatically discovered in our ICA workflow <https://github.com/SBRG/precise1k> based on regulon enrichments. These modules undergo statistical tests using the two-sided Fisher's exact test ($FDR < 10^{-5}$). Final module characterizations were determined through manual curation of the module's significant genes and a variety of annotation tools such as COG and GO terms. The statistical results mentioned above from automatic discovery can be found in the iModulon Table in the DatasetS2. The transcriptomic modules also were characterized in a similar way and have been thoroughly verified and validated in a series of publications with results that are now based on the 1035 profiles.

The only case where thresholding occurs for module correlations is when we want to compare the iModulon structures of the transcriptome to the proteome. See more details below in the response to bullet point 2.

2. From all *E. coli* genes, only 25 pairs are found that satisfy a PCC threshold of 0.25 (which is low to begin with). It is not clear why PCC and ICA would be the right methods to be applied in this problem, to begin with.

We thank the reviewer for mentioning this and allowing us the opportunity to clarify. 25 pairs of iModulons were found to satisfy the PCC threshold of 0.25, rather than 25 pairs of genes. The PCC threshold is set to 0.25 as the default threshold in the function used to compare iModulons between iModulon structures (`compare_ica` of the `pymodulon` package). While this may seem like a very low value, this is due to the non-uniformity of the gene-weight distributions for iModulons with similar functions across organisms or omics datatypes.

For instance, you can see two direct examples with Figure 3D and Figure 3E. The correlation for LeuO/Lrp is 0.65 while MetJ is 0.44, but it's very clear based on the gene enrichments (Figure 2H and 2I) that these iModulons are related to Methionine and Leucine.

Another example is Arginine and ArgR that have a correlation of 0.33, where both have genes like *argABCD*. The default threshold was identified and set after analyzing other species on iModulondb.org as part of our ongoing Modulome project that aims to analyze all publicly available high-quality transcriptomics

datasets using ICA. The main text has been edited to clarify this threshold selection.

With respect to your second comment, ICA works well in this case due to dimensionality reduction. Instead of having to correlate genes at an individual level, which has been done before, we have scaled this up. It's much easier to analyze 25-40 modules than 1000-4000 genes. PCC works better in this case than Spearman, for instance, since we care about the direct relationship between the raw gene weight values, rather than the monotonic relationship between variables.

3. The significance of the results (that are clearly stated in the abstract) is not clear. So modules in proteome/transcriptome have common genes, and modules at one layer are combination of another, which is to be expected and I am not sure what the lesson here is. The last result that "absolute proteome allocation can be inferred from the transcriptome alone", would be very impactful and amazing but I don't see support in this paper for accurate inference of proteome from transcriptome, piece-wise or otherwise.

As stated earlier, there were 4 main findings of the paper. The first two as you correctly stated was expected but still are important results as it has expanded the scope of what ICA can successfully accomplish. Additionally, it lays the foundation for the third and fourth results. We believe the third result is substantial as it shows how a top-down, quantitative/statistical approach (ICA) can lead to mechanistic interpretations of the relationships between two omics data types (in this case regulatory mechanisms). Similar to the reviewer, we believe the fourth result is the most impactful, especially because we were able to provide clear results with very modest statistical strength. We understand that the reviewer would prefer additional holdouts to better support our results but we were limited by the number of matched datasets (see bullet point 5).

4. Statistical support is generally limited throughout the claims - e.g. for interpretation of DIMA plots, or putative regulatory mechanisms based on matched modules.

We thank the reviewer for mentioning this and would like to highlight again the need for more matched datasets to allow for robust statistical analysis. As mentioned before, it took multiple years to generate the current matched dataset, and believe the results shown provide strong motivation to continue generating matched datasets to enable more rigorous statistical analysis.

5. The paper uses different regression models to showcase the predictability of proteome allocation using ti-Modulon activities. The authors provide results using leave-one-out-cross-validation while also making sure that replicates of the same

samples are not in the various folds. It is not clear what is the size of the validation set and whether a holdout was used for model selection.

Only 57 data points remain after proteome mass fractions and tiModulon activities were averaged for replicates. Due to this small dataset, we opted for leave-one-out-cross-validation (LOOCV) rather than k-fold-cross-validation or just a basic hold-out. Additionally, because LOOCV is rigorous and uses all available data for both training and validation, it can provide a more accurate estimate of model performance than the alternatives. Doing an additional holdout on top of LOOCV would make the dataset smaller and this is not desirable given the variance in some tiModulon activity distributions. As we generate more matched datasets, our predictions should improve as we can include additional holdout sets as you suggested. The discussion has been edited to mention this limitation and to highlight that our method is currently in its infancy, but still shows results worthy of follow-up studies.

Minor comments:

1. The authors distinguish genes as having a strong correlation if the corresponding regression models have $R^2 > 0.3$. Again the justification of that threshold is missing. The authors state that strong-correlation genes significantly differ from weak-correlation genes, but no p-value is given. Therefore, it is not clear what is the essential difference between strong and weak correlation genes and their impact on regression models. It will be interesting to analyze what is the biological reasons for their distinguishment.

We thank the reviewer for mentioning this. The adjusted R^2 0.3 cutoff was selected as a corrected threshold to use between exponential, broken line, and linear regression fits. An adjusted R^2 of 0.3 is equivalent to an R^2 value of 0.7 for linear regression in this dataset. This justification has been added to the main text.

We have changed the wording to remove significantly since a p value and statistical test is not given. The difference between the models that have strong correlations with their protein allocation and those that do not, comes down to the generalizability of the regression models which is reflected in the adjusted R^2 after LOOCV.

2. Although the PRECISE datasets (with identified 92 independent modules) are thoroughly verified and published, much more modules (201 independent modules) are identified in the transcriptome dataset (PRECISE1K) used in this study and the explanation of the different amount of identified independent modules is limited. In addition, the PRECISE1K dataset is not peer-reviewed.

PRECISE1k is our newest and most updated version, which has built upon previous versions of PRECISE beginning with the one you mentioned. Each iteration can be found at imodulondb.org under '*E. coli*'. Since then, the quality control has become more rigorous (replicates need higher correlations) and there are more datasets, hence the increase in independent components. The current PRECISE1k manuscript has been submitted, revised, resubmitted, and is under consideration at Cell Reports.

3. There are some inconsistencies (iModulon and i-Modulon) and I found it difficult to follow the decomposition results of the subset (Supplementary Data 3) as there is no explanation of what the resulting 60 components are and the relationship between these components and 41 components extracted from the entire dataset.

'i-Modulons' were used in previous works, but have recently been standardized to 'iModulon' the only times we use i-Modulon in this manuscript is when combining ti- and pi-Modulons for formatting.

We apologize for the inconvenience and appreciate the reviewer for noticing this. We have renamed the components in the dataset. They should now match up with their pi-Modulon matches.

4. It would be great to provide observations regarding matching between ti- and pi-modules, or the curve fitting results which are in contrast to existing knowledge.

We thank the reviewer for the suggestion and can say we have tried this. There are no ti- and pi-Modulon relationships we find that are in direct contrast to existing literature.

Reviewers' Comments:

Reviewer #1:

Remarks to the Author:

All my comments were addressed.

Reviewer #2:

Remarks to the Author:

The authors attended my comments and the paper has improved.

Reviewer #3:

Remarks to the Author:

I would like to thank the authors for their responses. Some of my comments regarding statistical analysis and the significance of the results remain, and I would like to have seen additional analysis that would support the robustness of the results and methods. I understand that "it took 2 years" to create the dataset, but the review is based on results and impact, rather than time spent.

Reviewers' Comments: **Black Text**

Author's Planned Response: **Blue Text**

Reviewer #3 (Remarks to the Author):

I would like to thank the authors for their responses. Some of my comments regarding statistical analysis and the significance of the results remain, and I would like to have seen additional analysis that would support the robustness of the results and methods. I understand that "it took 2 years" to create the dataset, but the review is based on results and impact, rather than time spent.

We apologize to the reviewer for not sufficiently addressing all of your comments previously. Based on your previous comments and this one, it seems the main point remaining is the lack of statistical power due to the lack of a holdout set from before and that the reviewer wanted additional analysis to show the robustness of our proteome allocation prediction. In doing so, we would then strengthen our fifth and last result (shown in main text Figure 5) that absolute proteome allocation can be inferred from the transcriptome alone, and thus improving the significance of the results in your eyes.

In order to do this, we conducted leave-one-out-cross-validation (LOOCV) in addition to a train/test split for further model analysis and robustness, as the reviewer mentioned previously. Each iModulon has 57 data points after averaging replicates. Due to this small sample size, we opted for splitting percentages of 10% only up to 30% in 5% increments. LOOCV was carried out on the training set, with the final model parameters evaluated against the test set.

For instance, the 10% split would result in 6 test points (10% of 57 is 5.7) and 51 training points. LOOCV was conducted on the 51 training points with each sample being the validation set once. The final model after training would then use the test data for model evaluation.

Model results for iModulons with strong correlations can be seen in the new Supplementary Figure 5 and model results for iModulons with weak correlations can be seen in the new Supplementary Figure 6 (reposted sequentially below for your convenience).

As can be seen in Supplementary Figure 5 (the first image above), the regression models for iModulons that are strongly correlated with their proteome allocation are actually fairly robust even for 57 samples. The Adjusted R² values stay above the 0.3 threshold for almost all splitting percentages across iModulons except for a couple of extreme (>20%) splits (Panel A above). This result indicates that the models fit the data well.

Furthermore, the root mean square errors (RMSEs) for the strongly correlated iModulons are also tight with an average standard deviation of 0.000438 (Panel B above). This average standard deviation indicates that the model can accurately infer the proteome allocation for new data points. The scale of RMSE relates to the magnitude of proteome allocation, which is why the Cra, Methionine, and RpoH iModulons all have relatively high RMSEs (0.043367, 0.043133, and 0.033590 % of the proteome, respectively). These three iModulons are the three largest by proteome mass (see Figure 5D in main text) and thus vary the most, so seeing them stand out is expected.

Supplementary Figure 6 (the second image above) shows the results for iModulons that are weakly correlated with their proteome allocation. The Adjusted R² values consistently stay below 0.3, indicating that the models don't fit the data well (Panel A above). The average standard deviation for the RMSEs for the weakly correlated iModulons is 0.000465, which is more than the average standard deviation for strongly correlated iModulons (Panel B above). These iModulons are much smaller by proteome mass, so one would expect the standard deviation to be less since it should scale with the magnitude of proteome allocation (only Translation, Lrp, ArcA, and Glyoxylate are large by mass, but we unexpectedly also see other iModulons with high RMSEs, see Figure 5D in main text). These results indicate that not only do the models not fit the data well, they also don't predict new data well.

Again, we wanted to apologize to the reviewer for not sufficiently addressing all of your concerns the first time. We hope that this additional analysis will support the robustness of our results and methods in your eyes. We appreciate your time, consideration, and especially your comments as we believe they have greatly improved the paper. We still believe that the regression models should be trained on all of the data since 57 data points is small (which is why we opt for LOOCV). However, the inclusion of additional holdouts didn't significantly deteriorate the results which improves our confidence in predicting proteome allocation from the transcriptome alone. Thus, the more robust analysis you have suggested has strengthened our results, in turn improving the significance of our work. We have added short commentary to the results and discussion (page 14 and page 17) mentioning this additional analysis and its implications. Longer discussion and the figures can be found in the Supplementary Information.

Reviewers' Comments:

Reviewer #2:

None

Reviewer #3:

None